# Aerodynamic Analysis of Camber Morphing Airfoils in Transition via Computational Fluid Dynamics

**DOI:** 10.3390/biomimetics7020052

**Published:** 2022-04-22

**Authors:** Bruce W. Jo, Tuba Majid

**Affiliations:** Advanced Dynamics, Mechatronics and Collaborative Robotics (ADAMS) Laboratory, Department of Mechanical Engineering, State University of New York (SUNY), Korea, Stony Brook University, Incheon 21985, Korea; tuba.majid@stonybrook.edu

**Keywords:** CFD, computational fluid dynamics, camber morphing, airfoil, analytical and numerical, benchmark, transition

## Abstract

In this paper, the authors analyze an important but overlooked area, the aerodynamics of the variable camber morphing wing in transition, where 6% camber changes from 2% to 8% using the two airfoil configurations: NACA2410 and NACA8410. Many morphing works focus on analyzing the aerodynamics of a particular airfoil geometry or already morphed case. The authors mainly address "transitional" or "in-between" aerodynamics to understand the semantics of morphing in-flight and explore the linearity in the relationship when the camber rate is gradually changed. In general, morphing technologies are considered a new paradigm for next-generation aircraft designs with highly agile flight and control and a multidisciplinary optimal design process that enables aircraft to perform substantially better than current ones. Morphing aircraft adjust wing shapes conformally, promoting an enlarged flight envelope, enhanced performance, and higher energy sustainability. Whereas the recent advancement in manufacturing and material processing, composite and Smart materials has enabled the implementation of morphing wings, designing a morphing wing aircraft is more challenging than modern aircraft in terms of reliable numerical modeling and aerodynamic analysis. Hence, it is interesting to investigate modeling the transitional aerodynamics of morphing airfoils using a numerical analysis such as computational fluid dynamics. The result shows that the SST k-ω model with transition/curvature correction computes a reasonably accurate value than an analytical solution. Additionally, the $C_{L}$ is less sensitive to transition near the leading edge in airfoils. Therefore, as the camber rate changes or gradually increases, the aerodynamic behavior correspondingly changes linearly.

## 1. Introduction

The term morphing in aerospace refers to technologies that possibly enhance an aircraft’s performance via alternating wing geometry corresponding to optimized wing shapes [1]. Recent airplane designs adapt fixed wings that could be said to be optimized in a single flight mode at a given task but far from aerodynamically optimized in others [2]. Meanwhile, adaptive or morphing wings could be more energy efficient [1,3,4,5]. Biomimetics is an interdisciplinary field where materials and mechanisms resemble or mimic biological systems processes [6]. From biomimetic aspects, avian morphology inspires us in morphing wing aircraft designs and efficient flight control accordingly [5]. Morphing wings enable aircraft to optimize flight mode by changing wing shapes through active adjustment of wing shapes corresponding to external or internal flight conditions while flight [1,3,7,8,9,10]. Most works in morphing focus on the mechanisms concerning conventional mechanical design or Smart materials and their proof-of-concept stages and material/structure-based internal mechanisms, which are far from realization and test flight of morphing aircraft [11,12,13,14,15,16,17,18,19,20,21,22,23,24,25,26,27,28,29,30,31,32,33,34,35,36,37,38,39,40,41,42,43,44,45].

The study of fluid flow around a morphing wing is essential to understanding aircraft performance and energy efficiency [46]. The aerodynamic loadings on the wing rely on the flow pattern. Also, it is coupled with the current shape of the wing [47,48,49,50,51]. A lifting-line theory could be used for well-defined airfoil shapes to analytically estimate and determine the L and D of the entire wing along the span direction [52]. Other standard techniques to study aerodynamics around the airfoils are experimental and numerical [53]. The significant advantage of experiments or testing is its accuracy; however, expensive in time and cost. For the case of a morphing wing, it would require replicating and controlling the behavior of a morphing wing inside an experimental apparatus such as a wind tunnel.

On the other hand, the numerical analysis offers flexibility in investigating various morphing geometries, providing the software’s accurate boundary conditions and settings [54]. This article’s related research works have numerical and experimental investigations [13,17,28,39,44]. However, it is necessary to eventually develop highly accurate and computationally inexpensive modeling and analysis tools to address morphing in transition states during flight to overcome the shortcomings of typical and conventional approaches and procedures [55]. As the first but most crucial step upward, authors investigate comparatively the numerical analysis of variable camber morphing airfoils and their behaviors in a transitional mode. In this present study, the authors have employed a CFD technique to establish the settings and conditions of airfoil models and adopted the National Advisory Committee for Aeronautics (NACA) [56]. First, the NACA2410 and NACA8410 are selected and analyzed where the entire wing has a taper ratio of one or a rectangular shape. Then, the results are compared with analytical results to validate them. Then, the NACA2410 on one side and the NACA8410 on the other side are modeled and studied to estimate their aerodynamics’ transitional and linear/nonlinear behaviors morphing airfoils.

## 2. Methodology

ANSYS FLUENT [54] has been adopted to analyze the aerodynamics of morphing airfoils of interest. There are a lot of numerical methods that can be used to study the aerodynamics of an airfoil within ANSYS FLUENT. However, as the simulated flow is laminar to turbulent transitional flow, a RANS turbulence model is used. The objective is to compute the associated stresses, RSM, Nonlinear eddy viscosity models, and Linear eddy viscosity models [46,56]. RSM, also known as an RST model, is a higher-level and elaborate turbulence model [57]. The eddy viscosity approach [58] was neglected, and the Reynolds stresses were computed directly. The exact Reynolds stress transport equation was related to the directional effects of the Reynolds stress fields. The nonlinear eddy viscosity models are a class of turbulence models, and an eddy viscosity coefficient relates the mean turbulence field to the mean velocity field in their nonlinearity. Nevertheless, as obtained from the RANS equation, the Reynolds stresses in this study are assumed in a linear constitutive relation with the mean flow straining field, which is what linear eddy viscosity models were designed.

Among various subcategories in linear eddy-viscosity models, the SST *k*−ω turbulence model—one of the best airfoil aerodynamics studies for laminar to turbulent transitional flow—was adopted. This model also predicts the adverse pressure gradients and separates flow. Furthermore, the model includes two additional equations to represent the turbulent flow properties that could account for historical effects, such as the convection and diffusion in turbulent energy. The turbulent kinetic energy, k, that determines the energy in the turbulent flows is the first variable. The second one is ω the specific turbulent dissipation that determines the scale of the turbulence.

The second-order upwind scheme is performed to discretize all spatial terms in conservation equations. The solver handles the pressure-velocity coupled algorithm, and the least-squares cell-based setting calculates the gradient term. Authors first solve three momentum equations and pressure-correction continuity equations, then calculate scalar values such as temperature and turbulence quantities. The residual criteria below $10^{-5}$ for all equations was used for convergence, and under-relaxation factors for each equation influence the process.

## 3. Modeling Approach and Mathematical Background

### 3.1. Assumption

The flow conditions for the analytical and CFD setup settings are set as follows:
**Air Properties**Density: 1$.25\;\mathrm{kg}/m^{3}$Viscosity *μ*: $1.6323\times 10^{-5}\;\frac{N.\;s}{m^{2}}$Far-field pressure: 70 kPa(abs)



**Geometry Properties**
Some of the geometrical explanation for the airfoil is shown in Figure 1 below.Wingspan: 9.89 mChord length: 0.54 mRe: 1,000,000where $Re=\frac{\rho V_{\infty }c\;}{\mu }$


ρ,$\;V_{\infty }$, c, and μ represent the density, free stream velocity, chord length, and molecular viscosity, respectively. So, the flow speed from the previous equation will be 24.68 m/s.

### 3.2. Geometry

An asymmetric airfoil NACA2410 for the baseline and NACA8410 for entirely morphed camber were used. In addition, the authors used MATLAB to generate the geometric profiles of airfoils for analysis in the ANSYS Design Modeler.

Figure 2 shows the basic geometry (NACA2410) of a morphing wing structure without any morphing. However, Figure 3 shows an entirely morphed on one end, no morphing on the other end NACA8410 and NACA2410.

#### 3.2.1. NACA Four-Digit Airfoil Specification

This NACA airfoil is characterized by four-digit NACA MPXX, which determines the camber, maximum camber position, and airfoil thickness. *M* is the number of the maximum camber divided by 100, and *P* is the number of the position of the maximum camber divided by 10. The remaining two digits denoted XX, indicate the thickness divided by 100. For instance, NACA2412, where *M* = 2, where camber is 0.02 or 2% of the chord, *P* = 4, the maximum camber is positioned at 0.4 or 40% of the chord, and XX = 12, the thickness is 0.12 or 12% of the chord [56]. The equations for camber geometry generation are tabulated in Table 1.

The equation gives the thickness distribution:(1)$$y_{t}=\frac{T}{0.2}\left(a_{0}x^{0.5}+a_{1}x+a_{2}x^{2}+a_{3}x^{3}+a_{4}x^{4}\right)$$
where $a_{0}=0.2969$, $a_{1}=-0.126$, $a_{2}=-0.3516$, $a_{3}=0.2843$, $a_{4}=-0.1015$, or −0.1035 for the closed trailing edge.

The constants $a_{0}$ to $a_{4}$ are a 20% thick airfoil and $\frac{T}{0.2}$ adjusts the constants to the desired thickness. At the trailing edge where *x* = 1, a finite thickness of 0.0021 chord width for a 20% airfoil exists. When a closed trailing edge is required, the $a_{4}$ could be adjusted correspondingly. The $y_{t}$ is a half thickness and applies to both sides in the camber line. Given x, it is straight-forward to calculate $y_{c}$ and the gradient and the thickness. The upper and lower surface position is perpendicular to the camber line.
(2)$$\theta =\mathrm{atan}\left(\frac{dy_{c}}{dx}\right)$$

The most efficient way to plot the airfoil is to iterate through equally spaced values in x and the upper and lower surface coordinates. The points are more widely spaced near the leading edge. The flatter sections can be seen on the plots. A cosine is used with uniform increments of β to group the points near the airfoil’s ends.
$$x=\frac{1-\cos\left(\beta \right)}{2}$$
where 0≤β≤π.

The thickness description curves for airfoil description are summarized in Table 2.

#### 3.2.2. Meshing

The far-domain is eight chord lengths away from the airfoil boundary, and a Design Modeler [54] adopted a C-type mesh. Then, a refined mesh is generated and applied next to the wall to capture details of *y*^+^ close to 1. Last, the distance between a wall in the airfoil and the first layer, y is calculated as below:(3)$$y=\frac{y^{+}\mu }{U_{\tau }\rho }$$
(4)$$U_{\tau }=\sqrt{\frac{\tau_{w}}{\rho }}$$
wall shear stress $\tau_{w}$ counted by:(5)$$\tau_{w}=\frac{1}{2}C_{f}\rho v_{air}^{2}$$

A literature survey refers to a formula for *Cf,* skin friction on a plate thus:(6)$$Cf=0.058Re^{-0.2}$$y value can be calculated by taking Re, μ, $v_{air}$ and ρ values and substituting them in previous equations.
$$Cf=36.6*10^{-4}\;U_{\tau }=\sqrt{0.5\left(36.6*10^{-4}\right){\left(24.68\right)}^{2}}=1.365\;m/s\;y=\frac{y^{+}\mu }{U_{\tau }\rho }=1.89\times 10^{-2}\;\mathrm{mm}$$
$$U_{\tau }=\sqrt{0.5\left(36.6*10^{-4}\right){\left(24.68\right)}^{2}}=1.365\;m/s$$
$$y=\frac{y^{+}\mu }{U_{\tau }\rho }=1.89\times 10^{-2}\;\mathrm{mm}$$


A Mesh Independence study was conducted as well. Richardson Extrapolation was adopted to calculate the grid convergence index, and it was confirmed that an index of less than 3% was obtained for the various grids tested. Computational time also plays a role in deciding the mesh element size. The grid comprising 7,060,448 elements (Element size: 0.01 m) was the most refined mesh tested, but as shown in Figure 4, varying the element size and thus the number of cells shows only minor changes in the values of $C_{D}$ we obtained from the simulations. Therefore, the element size was adjusted to 0.05 m for the rest of the study.

#### 3.2.3. Boundary Conditions

There are four boundary conditions imposed:(1)The inlet equals a velocity inlet,(2)The outlet equals a pressure outlet,(3)Faces with wing ends are symmetric, and(4)No-slip condition on the wall.

Computational models applied boundary conditions are shown in Figure 5a–e, as below.

## 4. Result

The k-ε (k-epsilon) model with the EWT (enhanced wall treatment), SST (shear stress transport) k-ω (k-omega), and SST k-ω with transition and curvature correction have been performed and analyzed. All models use a NACA2410 at 8° of AoA to verify the best model. The result shows the comparative study of $C_{L}$ and $C_{D}$ with analytical solutions as shown in Table 3. As a result, SST k-ω with IT/CC (intermittency transition/curvature correction) performs best in comparison. The result’s deviation in the analytical solution comes because the model presumes turbulent flow around the airfoil starts from the front of the leading edge. However, the flow around the airfoil surface is laminar in practice and similar to flow over a flat plate. Then, the flow changes to turbulent downstream of the airfoil. Since conventional turbulent models induce the turbulent $C_{D}$ for the entire airfoil, the simulation is larger than the experiment or analytics.

On the other hand, the SST k-ω with IT/CC outperforms in perspective. A detailed description is found in [59]. The TBL carries more energy and the $C_{D}$ is more significant than in the VBL. Meanwhile, the $C_{L}$ is less sensitive to the laminar and turbulence transition. Thus, the $C_{L}$ is higher in its value than experimental or analytical solutions [38].

### 4.1. NACA2410

The SST k-ω IT/CC is adopted to calculate the $C_{L}$ and $C_{D}$ where the NACA2410 3D airfoil with AoA changes from 0° to 8°, as shown in Table 4. Both $C_{D}$ and $C_{L}$ are compared with the analytical solutions, and the errors are below 13.8%.

The $C_{D}$ and $C_{L}$ are graphed while varying AoA as shown in Figure 6. The $C_{L}$ obtained from ANSYS FLUENT match the analytical solution’s most overlapping values. However, the $C_{D}$ from ANSYS is more significant than the one from the analytical solution. It is noted that the $C_{D}$ deviates from the analytical solution induced from the model setting and near-wall mesh resolution. Another reason for the deviation is that the analytical solution assumes the flow is laminar, but in reality, it is turbulent when *Re* = 1 million. So, the ANSYS values are correct. The variation is that the boundary layer in ANSYS is turbulent and of higher energy, which explains how the ANSYS values are more significant.

The pressure and velocity distribution show changes that travel far away from the wall, as shown in Figure 7 and Figure 8. The fluid domain that reaches up to 8 to 10 chord lengths seems large enough to capture the detail flows.

Figure 9 shows the contour of the turbulent kinetic energy, k for AoA of attacks of 1° and 8°. It is noted that turbulent energy is generated near the trailing edge and downstream. Thus, the mesh is recommended to be sufficiently refined near the downstream region.

The pressure distribution is shown in Figure 10 for AoA 1°, 5°, and 8°, respectively. As the AoA increases, it is noted that the pressure distribution at the top and bottom walls become separated from each. On average, both the $C_{D}$ and $C_{L}$ increase along with the AoA.

### 4.2. NACA8410

The simulations have been performed for the NACA8410 or 8% camber morphing case. The MATLAB generates a new mesh by applying the NACA8410 profiles at both ends. The same model setup and boundary conditions are applied for the NACA2410. The result of $C_{D}$ and $C_{L}$ for NACA8410 3D airfoil for the AoA varying from 0° to 8° is shown in Table 5.

The $C_{D}$ and $C_{L}$ of NACA8410 and NACA2410 are graphed while varying AoA as shown in Figure 11. It is noted that both configurations have a similar trend.

Figure 12 and Figure 13 show the pressure and velocity distribution of the NACA8410. Compared with Figure 7 and Figure 8 for the NACA2410, more significant pressure, and velocity variations are found, as shown in Figure 12 and Figure 13. The distribution stretches up to farther distances, similar to the $C_{D}$ and $C_{L}$ plotted in Figure 11. Increasing the camber rate leads to more significant in L and D as well.

The effects of turbulent kinetic energy, k are also explored by generating turbulent kinetic energy, k contours for the AoA of 1° and 8°, as shown in Figure 14. It is also noted that turbulent energy is mainly dense near the trailing edge and downstream in low and high AoA.

For a more direct and precise comparison, the pressure distribution plots around NACA2410 and NACA8410 airfoils are presented in Figure 15. 

The pressure difference between the top and the bottom surface of the NACA8410 shown in purple is more significant than the pressure differences of the NACA2410 in orange. That correlates to higher $C_{D}$ and $C_{L}$ for NACA8410 than for NACA2410.

### 4.3. From NACA2410 to NACA8410 Transition

The NACA2410 on one side and NACA8410 on the other have been analyzed to simulate morphing wings in transition. As a result, the cross-sectional geometry is linearly varied in between. 

The computational methodology was compared against analytical data and validated. Similar to the previous step, the same boundary conditions and settings were also applied to this case. Simulations were run for NACA2410 to 8410 transition 3D morphing wing geometry over a range of AoA from 0° to 8°. The $C_{D}$ and $C_{L}$ results are summarized in Table 6.

Figure 16, Figure 17, Figure 18 and Figure 19 show the NACA2410-NACA8410 transitional morphing scenario in their multi-axial input/output characteristics on deformation and stress.

The $C_{D}$ and $C_{L}$ vs. the AoA plots of NACA8410, NACA2410, and NACA2410 to 8410 transition cases are shown in Figure 16. As shown in Figure 20, the plot for the $C_{D}$ and $C_{L}$ of the transitional case (NACA2410 to NACA8410) lies in between the *L* plots for NACA2410 and NACA8410 airfoils. Similar trends are seen in the plots for the $C_{D}$. Whereas the $C_{L}$ curve for the transitional case lies in the middle of the NACA2410 and NACA8410 $C_{L}$ curves, the $C_{D}$ curve for the transitional case lies closer to NACA8410 $C_{D}$ curve, which could be interpreted as the $C_{L}$ being linear overall in the wing spar direction, the $C_{D}$ is more sensitive to structural deformation from coupling. In addition, since the overall trend is maintained over the wing span direction, $C_{L}$ and $C_{D}$ can be interpolated in-between. The pressure and velocity distribution of the transitional case with NACA8410 configuration on one side and NACA2410 configuration on the other side are shown in Figure 21 and Figure 22. A slightly more significant pressure difference is observed for the NACA8410 configuration end than for the NACA2410 configuration end, making sense as an 8% cambered airfoil configuration should generate a more significant pressure difference between the top and the bottom surfaces.

Figure 23 shows the contour of the turbulent kinetic energy, k at 5° AoA. Again, it is noted that the turbulent energy is highly dense near the trailing edge and downstream.

## 5. Summary

This paper presents an aerodynamic analysis of variable camber morphing wings via the CFD model with ANSYS FLUENT. The main focus lies on the transitional camber rate to resemble the camber morphing during flight. A refined mesh near the airfoil wall simulates the flow details. These suggested models perform based on the pre-defined initial and boundary conditions around a morphed wing. The results of the NACA2410, while varying AoA, are benchmarked and compared with analytical ones. The SST k-ω turbulence model with transition and curvature correction features has been adopted. It is noted that the $C_{L}$ is less sensitive to the flow transition near the airfoil’s leading edge. The $C_{D}$ and $C_{L}$ of NACA8410 airfoil are also calculated using the benchmarked ANSYS FLUENT setup. Lastly, a variable camber wing with end configurations matching the geometric profile of the NACA2410 airfoil on one side and the NACA8410 airfoil on the other side were analyzed to simulate the behavior of a camber morphing wing in transition. The $C_{D}$ and $C_{L}$ obtained from simulations are studied and analyzed. Furthermore, some essential flow parameters are graphed and analyzed for all cases, including that the flow characteristics change far from the airfoil wall and downstream. Therefore, it is suggested that fine mesh be considered to capture all the details. The significant contribution of this study is to explore and suggest a methodological approach to understand an aerodynamic analysis of morphing wings in flight more accurately.

## 6. Conclusions

A study of a numerical model and analysis of the aerodynamics of a variable camber morphing wing is presented in this paper. Contrary to a conventional wing, where the shape of the airfoil or wing is fixed, and the aerodynamic parameters can be computed for the fixed geometry, a morphing wing changes shape during the flight, which makes the task of computing the aerodynamic parameters of a shape-changing wing in transition challenging. Morphing Wing design concepts proposed morphing the wing geometry to a specific configuration at one end and another at the other end. However, aerodynamic analysis computing the $C_{L}$ and $C_{D}$ for such a morphing wing transition is not readily available. Hence, the authors in this paper present methodologies for transitional aerodynamics in a morphing wing by adopting a geometry; NACA2410 on one side; the NACA8410 on the other. At the same time, the intermediate geometry linearly transforms in the middle of the airfoil. The SST k-ω turbulence model with intermittency transition/curvature correction was used to compute the $C_{L}$ and $C_{D}$ for the morphing condition. The computational model used is validated against the analytical model, and an assuring benchmark was obtained with an error of less than 13.8% for the $C_{D}$ and 0.5% for $C_{L}$. Plots of $C_{L}$ and $C_{D}$ obtained for NACA2410, NACA8410, and NACA2410 to 8410 transition cases are compared. The results obtained for the morphing in transition case show that the variable camber morphing wing still performs better performance than the conventional airfoil. This study also indicates that morphing wings is a promising alternative to achieving a high-efficiency level on traditional aircraft. More importantly, it presents a method to analyze the aerodynamic performance of a scenario of a morphing wing in a transitional mode where a gradually changing morphing rate has been applied to a single body wing along the spanwise direction.

## Figures and Tables

**Figure 1 biomimetics-07-00052-f001:**
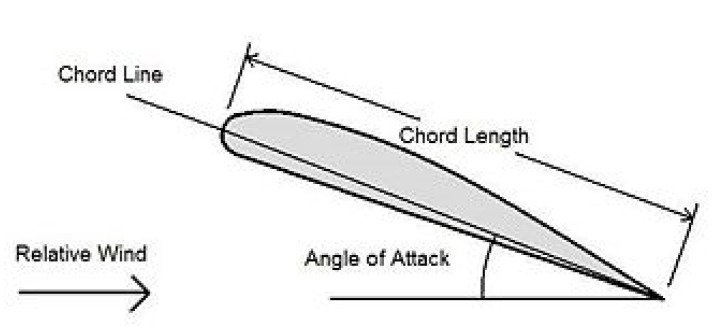
Cross-sectional geometry and description of an example airfoil [2].

**Figure 2 biomimetics-07-00052-f002:**
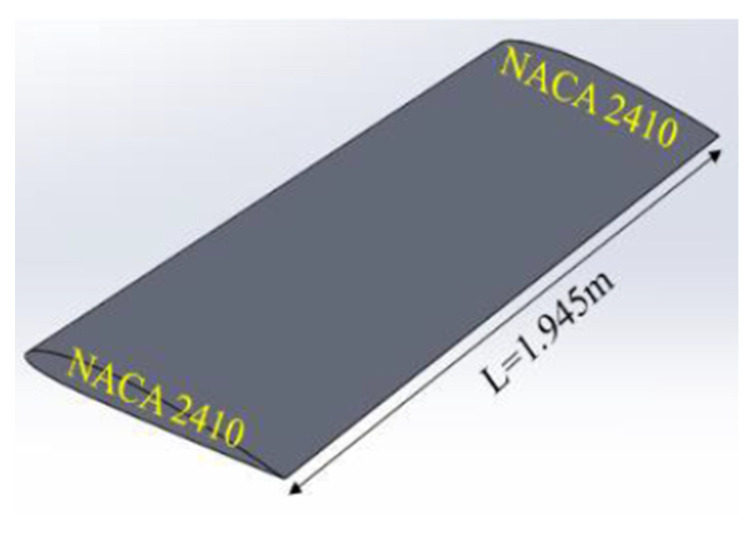
NACA2410-2410 wing and its dimensions.

**Figure 3 biomimetics-07-00052-f003:**
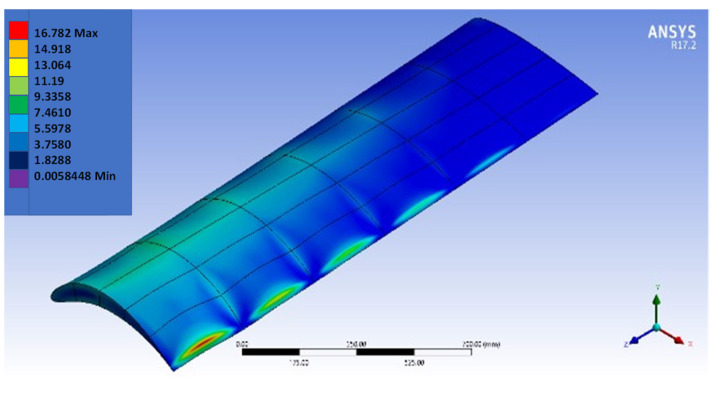
NACA8410 (at the wing tip) to NACA2410 (at the fuselage) wing under stress model.

**Figure 4 biomimetics-07-00052-f004:**
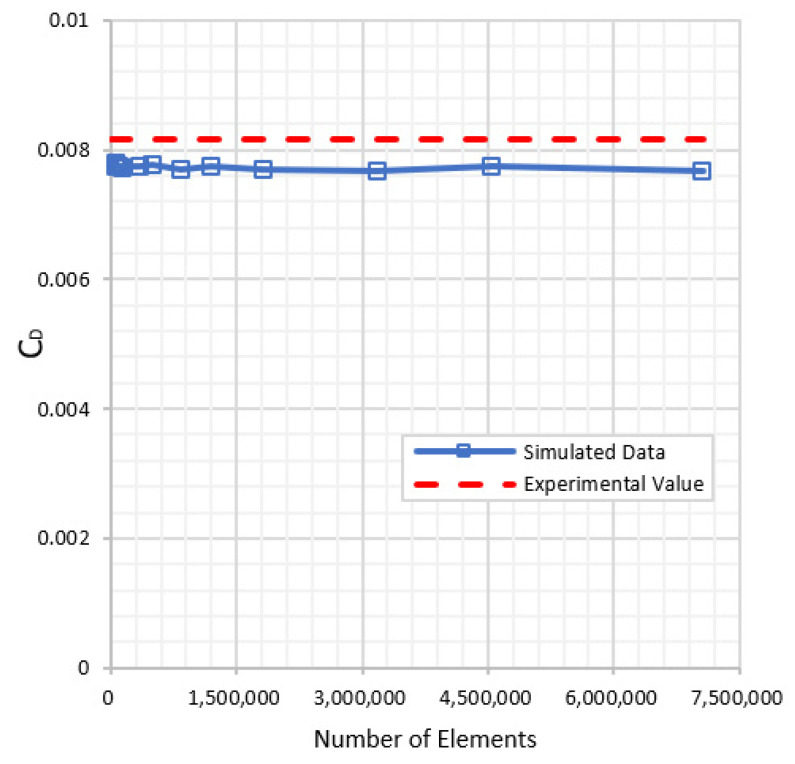
Mesh Independence Study (NACA Airfoil, Re = 6 Million, α = 0°).

**Figure 5 biomimetics-07-00052-f005:**
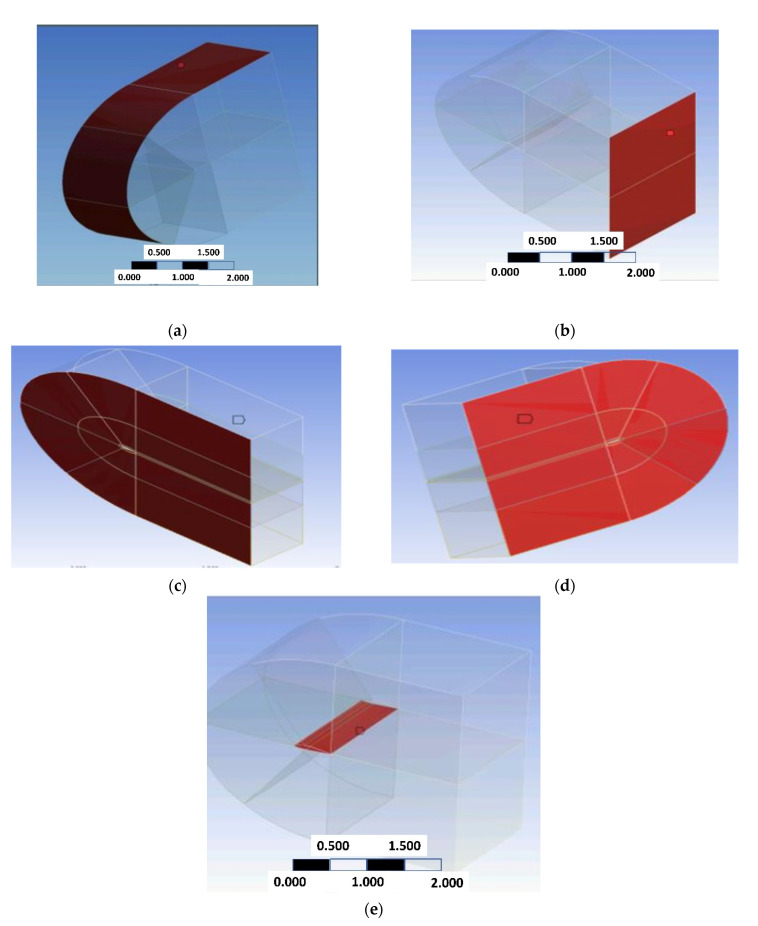
(**a**) input face and (**b**) outlet face. (**c**) the left-side symmetry faces and (**d**) the right-side symmetry face. (**e**) Airfoil inside the computational domain.

**Figure 6 biomimetics-07-00052-f006:**
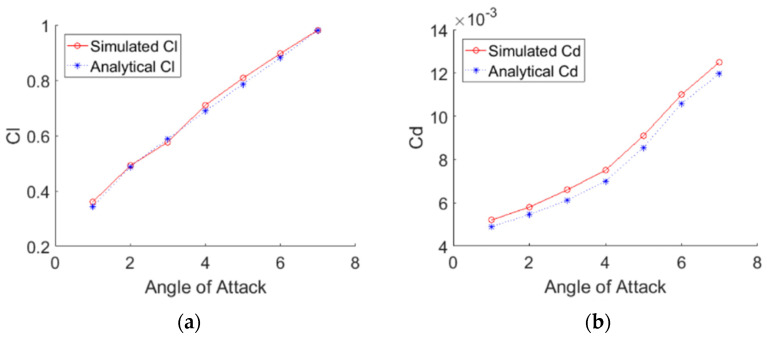
(**a**) $C_{L}$ and (**b**) $C_{D}$ vs. varying the AoA.

**Figure 7 biomimetics-07-00052-f007:**
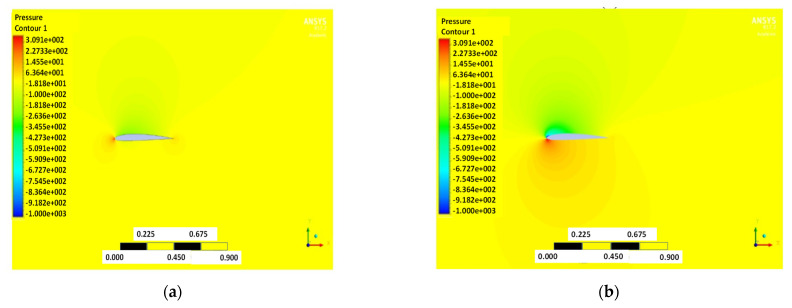
Pressure contour of NACA2410 with AoA (**a**) 1° and (**b**) 8°.

**Figure 8 biomimetics-07-00052-f008:**
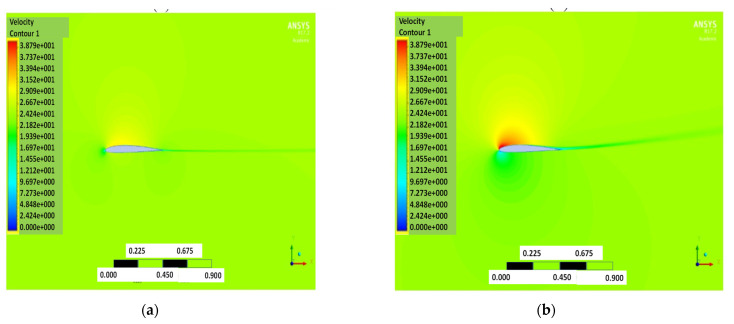
Velocity contour of NACA2410 with AoA (**a**) 1° and (**b**) 8°.

**Figure 9 biomimetics-07-00052-f009:**
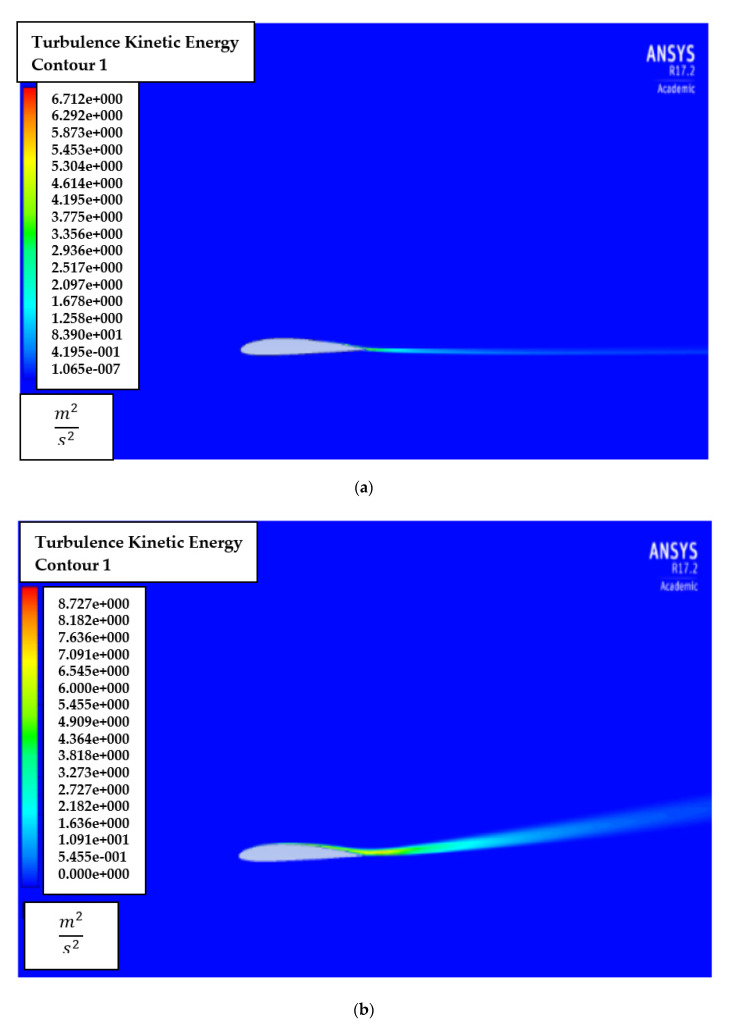
Turbulence intensity contour of NACA2410 with AoA (**a**) 1° and (**b**) 8°.

**Figure 10 biomimetics-07-00052-f010:**
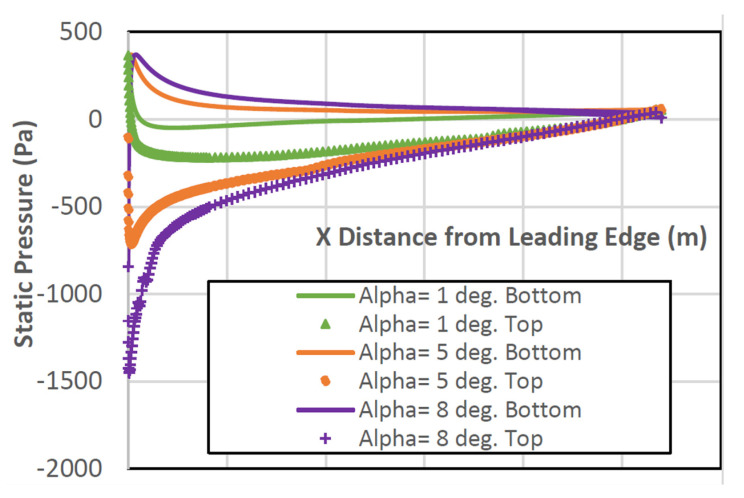
Pressure distribution around NACA2410 with varying AoA while x ticks from 0 with increment of 0.32; (0, 0.32, 0.64, 0.96, 1.28, 1.6, and 1.92, respectively).

**Figure 11 biomimetics-07-00052-f011:**
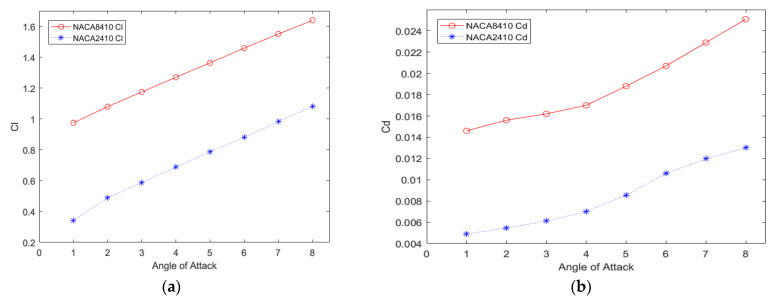
$C_{D}$ (**b**) and $C_{L}$ (**a**) of NACA2410 and NACA8410 vs. varying AoA.

**Figure 12 biomimetics-07-00052-f012:**
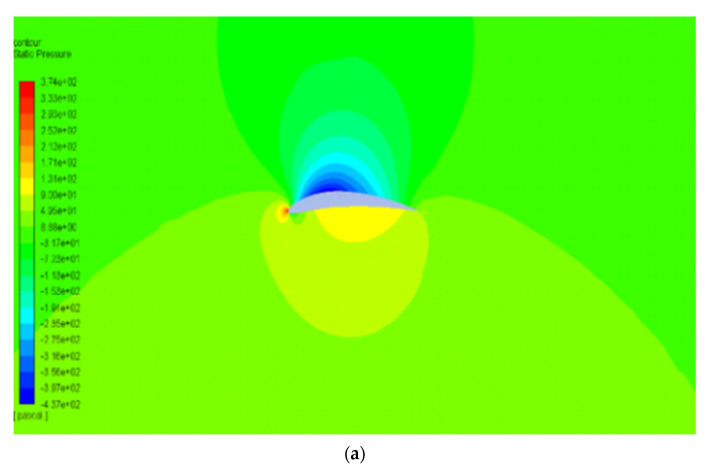
Pressure contour of NACA8410 with AoA (**a**) 1° and (**b**) 8°.

**Figure 13 biomimetics-07-00052-f013:**
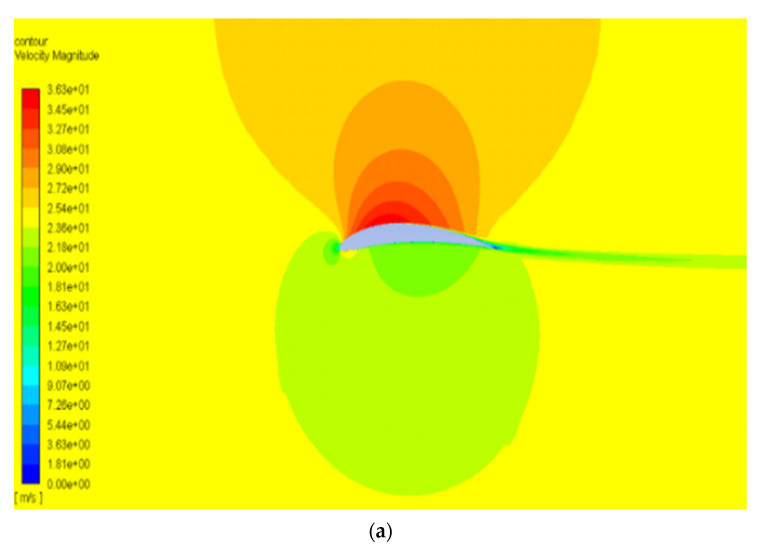
Velocity contour of NACA8410 with AoA (**a**) 1° and (**b**) 8°.

**Figure 14 biomimetics-07-00052-f014:**
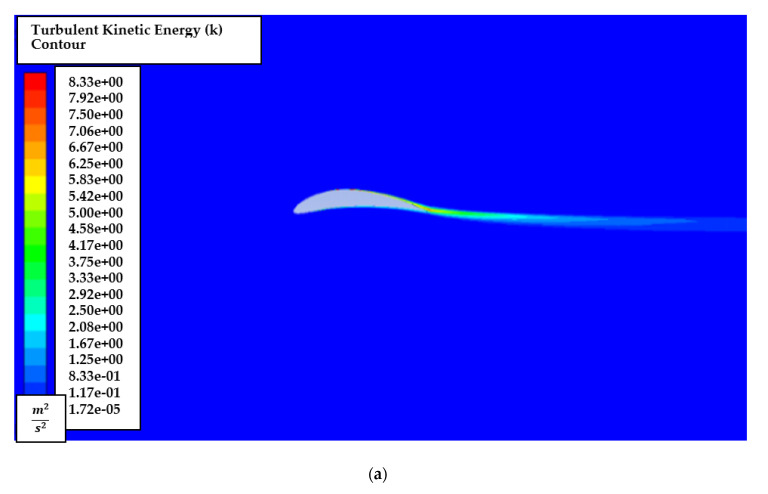
Turbulent kinetic energy, k contour of NACA8410 simulation with AoA (**a**) 1° and (**b**) 8°.

**Figure 15 biomimetics-07-00052-f015:**
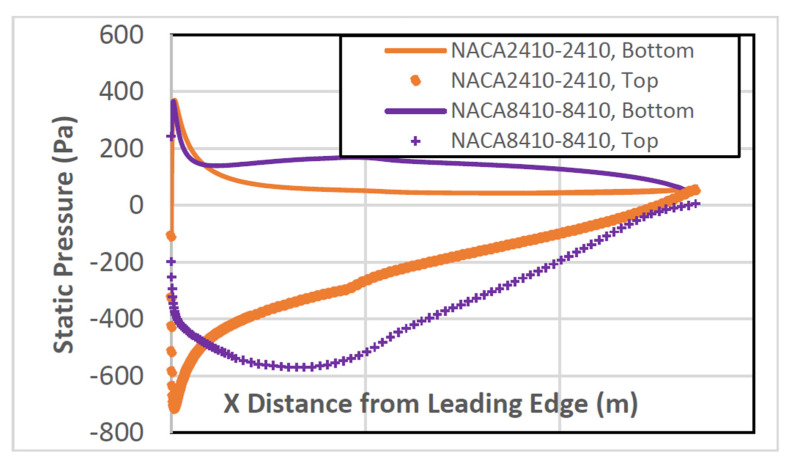
Pressure distribution of the NACA8410 with varying AoA while x ticks from 0 with increment of 0.32; (0, 0.32, 0.64, 0.96, 1.28, 1.6, and 1.92 respectively).

**Figure 16 biomimetics-07-00052-f016:**
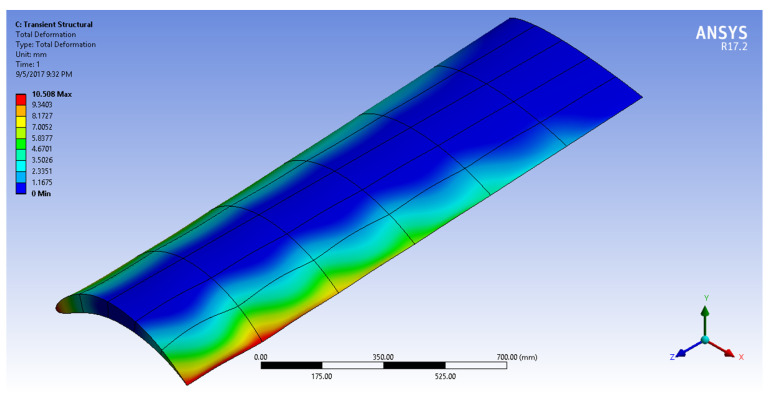
Input (deformation) distribution to morph NACA2410 to NACA8410 along the wing.

**Figure 17 biomimetics-07-00052-f017:**
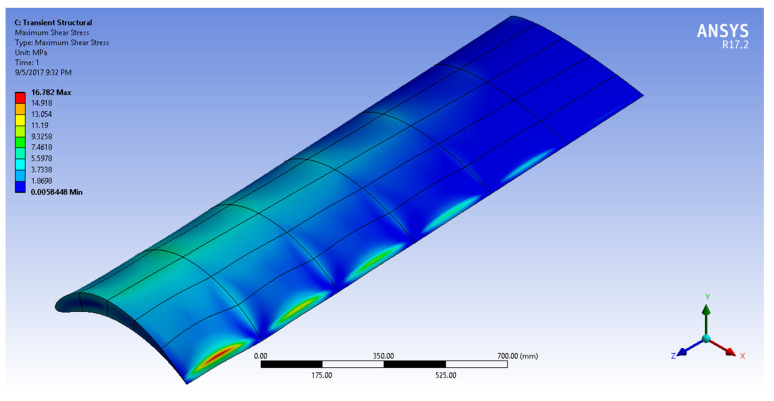
Output (maximum shear stress) distribution to morph NACA2410 to NACA8410 along the wing.

**Figure 18 biomimetics-07-00052-f018:**
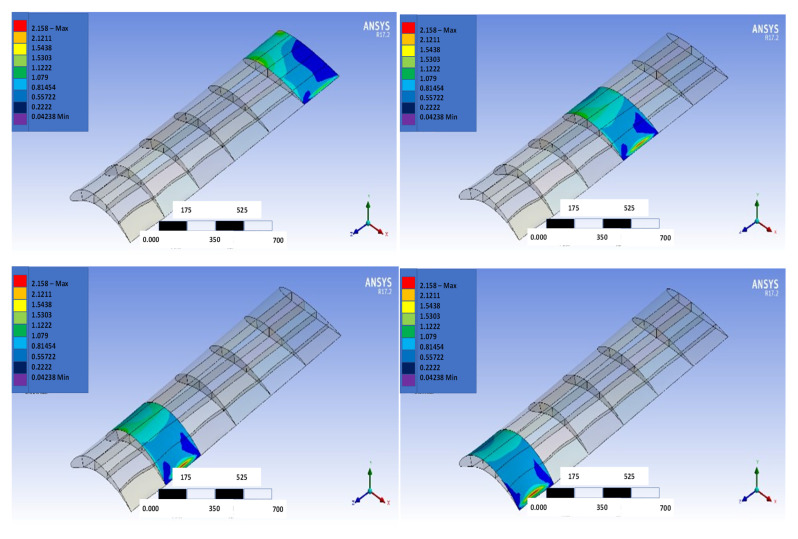
Sectional output (maximum shear stress) distribution to morph NACA2410 to NACA8410 along the wing.

**Figure 19 biomimetics-07-00052-f019:**
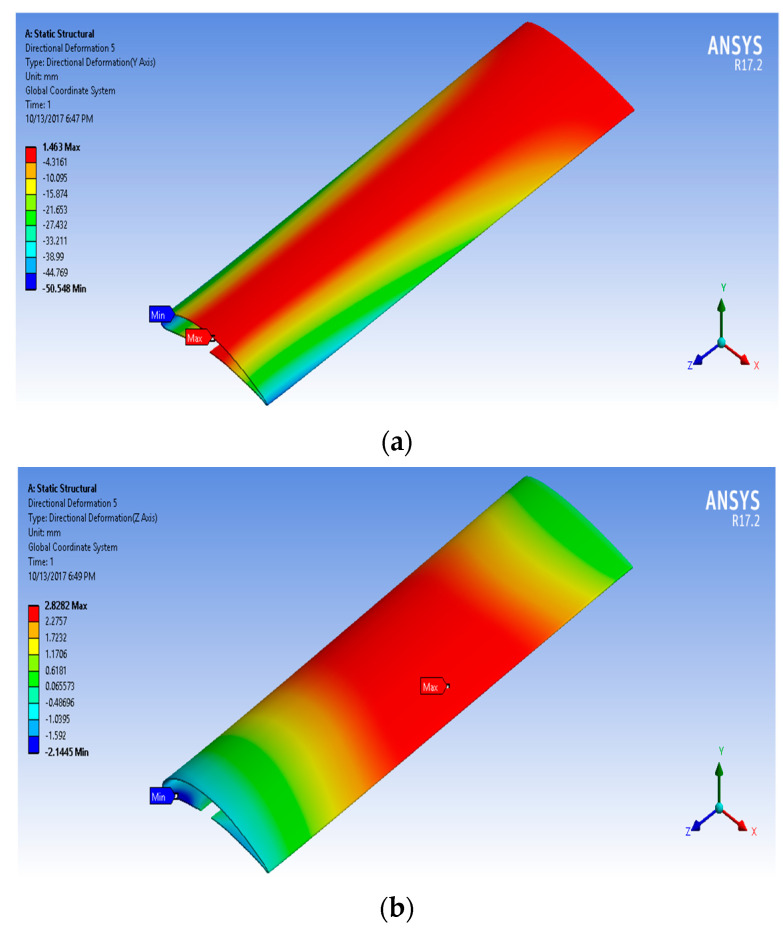
(**a**) Y axial output (deformation) distribution and (**b**) Z axial output (deformation) distribution) to morph NACA2410 to NACA8410 along the wing.

**Figure 20 biomimetics-07-00052-f020:**
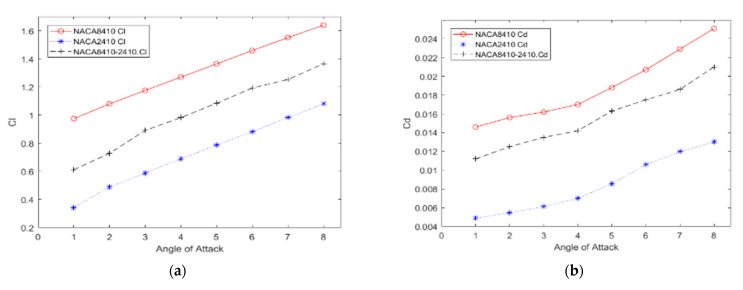
(**a**) $C_{L}$, (**b**) $C_{D}$ of NACA2410, NACA8410, & NACA2410 to NACA8410 transition vs. AoA.

**Figure 21 biomimetics-07-00052-f021:**
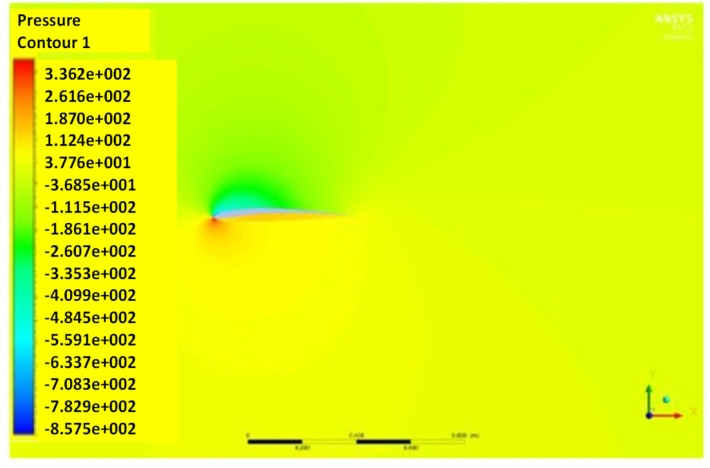
Pressure contour of the NACA2410 to NACA8410 transition at AoA 5°.

**Figure 22 biomimetics-07-00052-f022:**
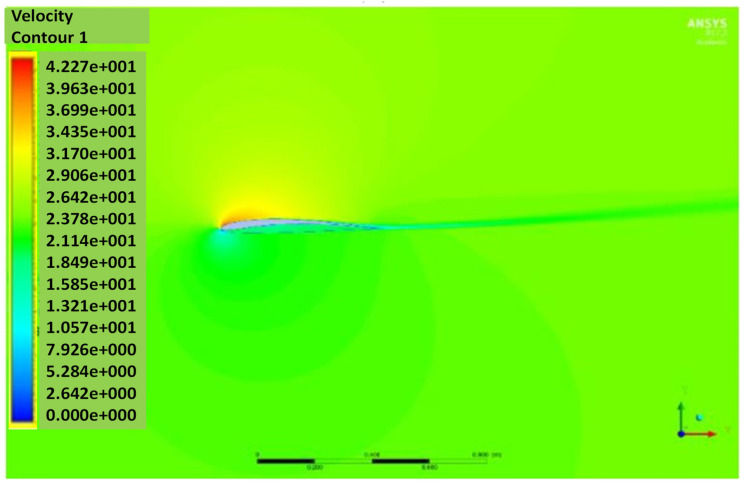
Velocity contour of NACA2410 to NACA8410 transition at AoA 5°.

**Figure 23 biomimetics-07-00052-f023:**
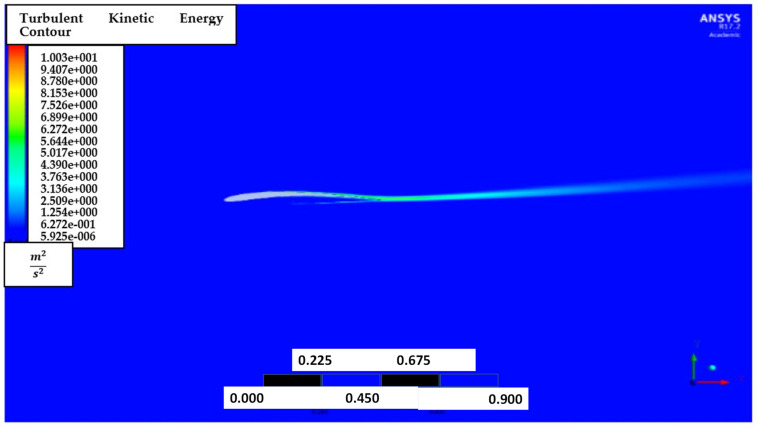
Turbulent kinetic energy, k contour of NACA2410 to NACA8410 transition at 5° AoA.

**Table 1 biomimetics-07-00052-t001:** Camber rate and gradient of geometry for airfoil description.

	Front (0≤x<p)	Back (p≤x≤1)
Camber Rate	$y_{c}=\frac{M}{p^{2}}\left(2Px-x^{2}\right)$	$y_{c}=\frac{M}{{\left(1-p\right)}^{2}}\left(1-2P+2Px-x^{2}\right)$
Gradient	$\frac{dy_{c}}{dx}=\frac{2M}{p^{2}}\left(P-x\right)$	$\frac{dy_{c}}{dx}=\frac{2M}{{\left(1-p\right)}^{2}}\left(P-x\right)$

**Table 2 biomimetics-07-00052-t002:** Thickness description curves for airfoil description.

Upper Surface	$x_{u}=x_{c}-y_{t}\sin\left(\theta \right)$	$y_{u}=y_{c}+y_{t}\cos\left(\theta \right)$
Lower Surface	$x_{l}=x_{c}+y_{t}\sin\left(\theta \right)$,	$y_{l}=y_{c}-y_{t}\cos\left(\theta \right)$

**Table 3 biomimetics-07-00052-t003:** Turbulence model effects on $C_{L}$ and $C_{D}$.

Model	A	Analytical $\;C_{L}$	Analytical $\;C_{D}$	Numerical $\;C_{L}$	Numerical $\;C_{D}$	$C_{L}$ Error (%)	$C_{D}$ Error (%)
Realizable k-ε, EWT	8	1.0807	0.0134	1.0800	0.0206	0.06	54
k-ω, SST	8	1.0807	0.0134	1.0810	0.0171	0.03	28
k-ω, SST, IT/CC	8	1.0807	0.0134	1.0750	0.0148	0.53	10

**Table 4 biomimetics-07-00052-t004:** Benchmarking of NACA2410 simulated results against analytical ones.

Model	α	Analytical $\;C_{L}$	Analytical $\;C_{D}$	Numerical $\;C_{L}$	Numerical $\;C_{D}$	$C_{L}$ Error (%)	$C_{D}$ Error (%)
k-ω, SST, IT/CC	1	0.343	0.005	0.361	0.0052	5.2	4.0
k-ω, SST, IT/CC	2	0.488	0.006	0.492	0.0058	0.8	3.3
k-ω, SST, IT/CC	3	0.588	0.006	0.577	0.0066	1.9	10.0
k-ω, SST, IT/CC	4	0.688	0.007	0.712	0.0075	3.5	7.1
k-ω, SST, IT/CC	5	0.786	0.009	0.808	0.0091	2.8	1.1
k-ω, SST, IT/CC	6	0.881	0.011	0.897	0.0110	1.8	0
k-ω, SST, IT/CC	7	0.982	0.012	0.982	0.0124	0	3.3
k-ω, SST, IT/CC	8	1.080	0.013	1.075	0.0148	0.5	13.8

**Table 5 biomimetics-07-00052-t005:** NACA8410 simulated results.

Model	α(AoA) °	Numerical $C_{L}$	Numerical $\;C_{D}$
k-ω, SST, IT/CC	1	0.9749	0.0146
k-ω, SST, IT/CC	2	1.0795	0.0156
k-ω, SST, IT/CC	3	1.1751	0.0162
k-ω, SST, IT/CC	4	1.2696	0.0170
k-ω, SST, IT/CC	5	1.3634	0.0188
k-ω, SST, IT/CC	6	1.4588	0.0207
k-ω, SST, IT/CC	7	1.5505	0.0229
k-ω, SST, IT/CC	8	1.6392	0.0251

**Table 6 biomimetics-07-00052-t006:** NACA2410-8410 end simulated results.

Model	AoA°	Numerical $\;C_{L}$	Numerical $\;C_{D}$
k-ω, SST, IT/CC	1	0.612	0.0112
k-ω, SST, IT/CC	2	0.728	0.0125
k-ω, SST, IT/CC	3	0.891	0.0135
k-ω, SST, IT/CC	4	0.982	0.0142
k-ω, SST, IT/CC	5	1.084	0.0163
k-ω, SST, IT/CC	6	1.192	0.0175
k-ω, SST, IT/CC	7	1.253	0.0186
k-ω, SST, IT/CC	8	1.365	0.021

## Data Availability

Not applicable.

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
