# Peer review of "Aerodynamic Analysis of Camber Morphing Airfoils in Transition via Computational Fluid Dynamics"

_biomimetics, 2022, doi:10.3390/biomimetics7020052_

Round 1

Reviewer 1 Report

In this paper, the authors tried to investigate the aerodynamics of morphing wings to utilize the airfoil of birds. It is a very interesting idea.
From my understanding, the morphing wing should have the same airfoil along the span direction and change the airfoil shape intentionally.
The interesting point will be the aerodynamics of the airfoil when the airfoil is changing dynamically.

The biggest problem of this paper was that the shape of the morphing wing was not clearly shown.
From fig 2, I GUESSED that your "morphing" wing means that one end has NACA 2410 airfoil and the other end has NACA8410 airfoil, and the airfoil shape changes along the span direction gradually.

If my guess is correct, the boundary condition at both ends should not be the symmetric condition. Because both ends have different airfoil shapes.
Additionally, if the airfoil shape changes along the span direction and it is a key to your results, you should show the static pressure distribution at different cross-sections (at the different span locations).
The obtained results for your morphing wing are almost the same as the average of NACA 2410 and NACA 8410. Then you should compare with NACA 5410 (mean camber value of two airfoils).

The followings are the figures that were not clearly shown.
The numbers in the color bar were not clear in fig 7,8,9,12,13,14,17,18, and 19.
The static pressure should be addressed as gauge pressure (not absolute pressure) in fig. 10, and 15.
Figure 10 and 15 does not include the number on X-axis.
Figures 17, 18, and 19 had different scales for the two figures and should be recreated.

Author Response

Dear reviewer,

Please see the attached response on your feedback and comments.

Authors greatly appreciate your efforts to improve this article.

Thank you

Reviewer 2 Report

The authors describe simulation results of non-morphing and morphing airfoils, comparing the results with/without morphing. the topic might be interest for readers. However, followings must be revised for publication.

(1)In 1.Introduction, there is no description about the relationship between biomimetics and morphing airfoil. The authors should state the relationship.

(2)In 2.Methodology, the explanation of models is too simple. Additional explanation should be useful for readers. For example, whose models were used as the RSM and Nonlinear eddy viscosity models?

(3)In 2.Methodology, there is a sentence "The second-order upwind scheme is performed for discretization for all spatial terms .....". However, generally, diffusion terms are not discretized by upwind scheme. Is this sentence correct?

(4)In 3.2.2.Meshing, there is no description on grid-independence study, grid structure, and grid number. The authors should express these information.

(5)In 4.Results, what are the terms: Real. k-e, EWT, IT/CC, and VBL? The authors should explain these terms in the text.

(5)About Fig.16, the authors states "..... the Lift coefficient curve for the transitional case lie somewhat in the middle of the NACA2410 and NACA 8410 Lift curves, the Drag coefficient curve for the transitional case lies closer to NACA8410 Drag coefficient curve". The author should describe the reason why this difference happens, investigating the detail of the flow behavior.   

(6)A lot of grammatical errors and mistypes are found. Please correct them.

-End-   

Author Response

Dear reviewer,

Please see the attached response on your feedback.

Authors greatly appreciate your efforts to improve this article.

Thank you.

Round 2

Reviewer 1 Report

Some figures should be improved as follows.
For Fig. 9 and 14, the color of letters should be white and the range of the color should be the same for two figures. 
The resolution of figures should be updated so that the letters in figures can be recognizable.

For Fig. 12 and 13, the resolution should be updated so that the letters in figures can be recognizable.
For Fig. 21 and 22, the dimension of the airfoils should be the same in two figures so that we can discuss the difference of the distributions.
For Fig. 24, the color of letters should be white and the resolution should be updated so that the letters in figures can be recognizable.

typo
line 178, NACA84120 > NACA8410

Author Response

Please see the attached response.

Reviewer 2 Report

None

Author Response

Authors greatly appreciate your feedback.
